# β-Ketoenamine Covalent Organic Frameworks—Effects of Functionalization on Pollutant Adsorption

**DOI:** 10.3390/polym14153096

**Published:** 2022-07-29

**Authors:** Tiago F. Machado, Filipa A. Santos, Rui F. P. Pereira, Verónica de Zea Bermudez, Artur J. M. Valente, M. Elisa Silva Serra, Dina Murtinho

**Affiliations:** 1CQC-IMS, Department of Chemistry, University of Coimbra, 3004-535 Coimbra, Portugal; tiago.f.machado@hotmail.com (T.F.M.); filipa_santos1997@hotmail.com (F.A.S.); melisa@ci.uc.pt (M.E.S.S.); dmurtinho@ci.uc.pt (D.M.); 2Chemistry Department and Chemistry Center, University of Minho, 4710-057 Braga, Portugal; rpereira@quimica.uminho.pt; 3Chemistry Department and CQ-VR, University of Trás-os-Montes e Alto Douro, 5000-801 Vila Real, Portugal; vbermude@utad.pt

**Keywords:** covalent organic frameworks, β-ketoenamine, adsorption, methylene blue, methyl orange, heavy metals

## Abstract

Water pollution due to global economic activity is one of the greatest environmental concerns, and many efforts are currently being made toward developing materials capable of selectively and efficiently removing pollutants and contaminants. A series of β-ketoenamine covalent organic frameworks (COFs) have been synthesized, by reacting 1,3,5-triformylphloroglucinol (TFP) with different C2-functionalized and nonfunctionalized diamines, in order to evaluate the influence of wall functionalization and pore size on the adsorption capacity toward dye and heavy metal pollutants. The obtained COFs were characterized by different techniques. The adsorption of methylene blue (MB), which was used as a model for the adsorption of pharmaceuticals and dyes, was initially evaluated. Adsorption studies showed that –NO_2_ and –SO_3_H functional groups were favorable for MB adsorption, with TpBd(SO_3_H)_2_-COF [100%], prepared between TFP and 4,4′-diamine- [1,1′-biphenyl]-2,2′-disulfonic acid, achieving the highest adsorption capacity (166 ± 13 mg g^−1^). The adsorption of anionic pollutants was less effective and decreased, in general, with the increase in –SO_3_H and –NO_2_ group content. The effect of ionic interactions on the COF performance was further assessed by carrying out adsorption experiments involving metal ions. Isotherms showed that nonfunctionalized and functionalized COFs were better described by the Langmuir and Freundlich sorption models, respectively, confirming the influence of functionalization on surface heterogeneity. Sorption kinetics experiments were better adjusted according to a second-order rate equation, confirming the existence of surface chemical interactions in the adsorption process. These results confirm the influence of selective COF functionalization on adsorption processes and the role of functional groups on the adsorption selectivity, thus clearly demonstrating the potential of this new class of materials in the efficient and selective capture and removal of pollutants in aqueous solutions.

## 1. Introduction

Uncontrolled pollution due to human activity and agricultural and industrial production is one of the main environmental concerns currently faced by the scientific community, and a tremendous challenge for water and soil preservation [1]. Amongst the pollutants typically found in contaminated waters, some new emerging classes of organic compounds such as volatile organic compounds (VOCs) or nonsteroidal noninflammatory drugs (NSAIDs) have raised increasing concern over the decades. Other organic pollutants, such as food additives, biotoxins, dyes, pesticides, and organic derivatives of cosmetic, hygiene, or pharmaceutical products, are also consistently found in contaminated waters and soils, as well as inorganic substances such as heavy metals or nonmetals, halides, cyanides, nitrates, or even radionuclides [2,3,4,5]. The vast majority of these compounds have disastrous effects on human health, biodiversity, and ecosystems, even at low concentrations. It is therefore crucial to find efficient, simple solutions to safely remove them and properly treat them [1,2,6]. Currently, several different techniques and processes are employed for the removal and degradation of contaminants from residual waters, such as coagulation, flocculation, sedimentation, chlorination, advanced oxidation processes, photochemical and electrochemical degradation, reverse osmosis, adsorption, or filtration [2].

Adsorption and filtration, in particular, are cheap, nonintrusive, sustainable approaches and, thus, very attractive solutions for water remediation. In this context, a wide range of porous materials has since been tested as adsorbents, namely, zeolites [7], nanosponges [8], inorganic porous materials [9], activated carbon [10,11,12], graphene and carbon nanomaterials [13,14,15], Metal–Organic Frameworks (MOFs) [1,6,16], as well as a variety of organic polymer classes, such as Porous Organic Polymers (POPs) [17], Porous Aromatic Frameworks (PAFs) [18], Polymers of Intrinsic Microporosity (PIMs) [19], and biopolymer blends [20,21]. Recently, COFs (Covalent Organic Frameworks) have been gaining considerable attention for the adsorption and removal of pollutants from water due to their intrinsic properties of chemical stability, high, permanent porosity, and building versatility and customizability. Thus, there have been many recent attempts to implement COFs as pollutant adsorbents in both real and simulated residual water samples [22].

COFs are generally built from rigid, geometrically defined organic building blocks, resulting in robust, covalently bonded crystalline networks that extend in 2 or 3 dimensions [23], with a wide range of potential applications, including gas storage [24], heterogeneous catalysis [25,26], chemical sensing [27,28], optoelectronics [29], and pollutant adsorption [2,22,30]. One of the main advantages of COF materials is the ability to easily introduce pore wall modifications (either pre- or post-synthesis) without affecting the structural integrity of the resulting solid. This means COFs can be customized and fine-tuned for selective, specific pollutant adsorption and separation [31]. In this respect, β-ketoenamine-type COFs are particularly attractive for adsorption in aqueous solution. They are synthesized by reacting symmetric diamines with symmetric polyaldehydes bearing hydroxyl groups in the ortho position, such as 1,3,5-triformylphloroglucinol (TFP). The imine to ketoenamine tautomerization is an irreversible process, highly resistant to hydrolysis, making the resulting COFs suitable for pollutant removal from water [32].

Several COFs have been proposed for the adsorption of heavy metals [33,34,35,36,37,38] and organic pollutants [39,40,41,42,43,44,45,46,47] from aqueous solution. Zhu et al. first developed a triazine-based COF, TS-COF-1, exhibiting a high methylene blue (MB) dye adsorption capacity (1.691 mg g^–1^), demonstrating the potential of this class of materials for water treatment [48]. Dang et al. reported a series of sulfonic acid (–SO_3_H)-functionalized β-ketoenamine COFs impregnated with ionic liquid derivatives, showing even higher MB adsorption (up to 2.865.3 mg g^–1^) [49].

In this paper, a variety of TFP-based β-ketoenamine COFs, with varying pore sizes and degrees of pore wall functionalization, were synthesized and their performance in adsorbing dyes and heavy metal pollutants, in aqueous solution, was assessed. The COF materials were prepared via microwave and solvothermal methods and characterized by scanning electron microscopy (SEM), X-ray diffraction (XRD), thermogravimetric analysis (TGA), and nitrogen adsorption–desorption analysis. The COFs were tested for the adsorption of MB and methyl orange (MO) pollutants in aqueous solution. Additionally, the adsorption ability of nitro (–NO_2_) and –SO_3_H-functionalized COFs for adsorption of the heavy metal ions Cu(II), Ni(II), Pb(II), and Cd(II) was evaluated. The information obtained from the characterization and adsorption results was compared in order to evaluate the influence and importance of the materials’ properties on their performance.

## 2. Materials and Methods

### 2.1. Reagents

Reagents and solvents were obtained from the following suppliers: phloroglucinol (anhydrous, 98%) and mesitylene (>98%) (Alfa Aesar, Karlsruhe, Germany); hexamethylenetetramine (HMTA, 99.5%) and methylene blue (MB, >99%) (Riedel-de Haën, Seelze, Germany); 4-phenylenediamine (Pa, 99.5%) (Sigma Aldrich, Schnelldorf, Germany); 4,4′-ethylenedianiline (Bba, 99%) (Fluka, Neu-Ulm, Germany); methyl orange (MO, >99%) (Merck, Darmstadt, Germany); benzidine dihydrochloride (B.D.H., England, UK); 2,2′-dinitro-[1,1′-biphenyl]-4,4′-diamine ([Bd(NO_2_)]_2_, 99%), 4,4′-diamine-[1,1′-biphenyl]-2,2′-disulfonic acid ([Bd(SO_3_H)]_2_, >70%), and trifluoroacetic acid (TFA, 99%) (Fluorochem, Hadfield, UK); sodium sulfate (Na_2_SO_4_, anhydrous, 99%+) and sodium hydroxide (>99%) (José Manuel Gomes dos Santos, Lisboa, Portugal); deuterated chloroform (99.8%) and deuterated dimethyl sulfoxide (99.8%) (Eurisotop, Saint-Aubin, France); sodium bicarbonate (>99%) (LabKem, Barcelona, Spain); nitric acid (65%) (Panreac, Barcelona, Spain); hydrochloric acid (37%)—Fisher Chemicals (Merelbeke, Belgium); glacial acetic acid (AcOH, >99%) (Chem-Lab, Zedelgem, Belgium). Heavy metal solutions were prepared using copper (II) nitrate hemipentahydrate (p.a., Chem-Lab, Zedelgem, Belgium), lead (II) nitrate (≥99.0%, Sigma-Aldrich, Schnelldorf, Germany), cadmium (II) nitrate tetrahydrate (≥99.0%, Sigma-Aldrich, Schnelldorf, Germany), and nickel (II) nitrate hexahydrate (crystals, Sigma-Aldrich, Schnelldorf, Germany). All other solvents were purified or dried prior to use following standard procedures. Other commercially available compounds were used without further purification.

### 2.2. Synthesis of Monomers and COF Materials

1,3,5-triformylphloroglucinol (TFP). The synthesis of TFP was adapted from the procedure reported by Chong et al. [50]. Phloroglucinol (3.0 g, 23.8 mmol) and HMTA (7.5 g, 52.8 mmol) were dissolved in TFA (45 mL) in a two-neck round-bottom flask. The mixture was refluxed, under inert atmosphere and magnetic stirring, for 2.5 h. Subsequently, under the same conditions, HCl 3 M (75 mL) was added dropwise, for approximately 1 h. After cooling, the reaction mixture was filtered through a celite bed, and the product was extracted with dichloromethane (4 × 75 mL) and chloroform (2 × 75 mL). The combined organic phases were dried with anhydrous Na_2_SO_4_ and filtered. After solvent evaporation, the product was washed with hot ethanol, resulting in a thin orange powder (12% yield). ^1^H RMN (CDCl_3_, 400 MHz): 10,16 (s, 3H); 14,12 (s, 3H).

Benzidine (Bd). Benzidine dihydrochloride (1.3 g, 5.0 mmol) was dissolved in distilled water (200 mL), under gentle heating and magnetic stirring. Upon dissolution, sodium hydroxide 20% (*m*/*v*) (2 mL) was added, and the resulting precipitate was filtered and dried. The neutralized amine was obtained as a brown solid.

COF Syntheses via Microwave method. Microwave-assisted COF synthesis was conducted according to the previously reported procedure [51]. TFP (0.12 g, 0.6 mmol, 1 Equation) and the respective diamine (0.9 mmol, 1.5 Equation) were added to a mesitylene/dioxane/acetic acid 3 M (3:3:1 *v/v/v*) mixture (14 mL), in a microwave tube. The mixture was sonicated (1 min), for homogenization and degassing, and subjected to microwave irradiation (300 W) at 100 °C under magnetic stirring for 1 h. Upon cooling, the COF precipitate was thoroughly washed with alternating portions of acetone and distilled water, until the absence of coloration of the filtrate.

COF Syntheses via Solvothermal method. Functionalized COFs were also prepared via solvothermal synthesis [52]. TFP (0.12 g, 0.6 mmol) and the respective diamine (combined molar quantity 0.9 mmol, 1.5 Equation) was added to a mesitylene/dioxane/acetic acid 3 M (3:3:1 *v*/*v*/*v*) mixture (7 mL) in a round-bottom flask. The mixture was sonicated (1 min), for homogenization and degassing, and then refluxed under inert atmosphere and magnetic stirring for 48 h. Upon cooling, the obtained solid was thoroughly washed with alternating portions of acetone and water until the absence of coloration of the filtrate.

Functionalized COFs were synthesized at different molar ratios, by mixing the functionalized diamine with its nonfunctionalized counterpart (Bd). The molar ratios used for preparing the various functionalized COFs are shown in Table 1.

For both synthetic procedures, COF materials were purified via the Soxhlet extraction method, first under tetrahydrofuran and then under acetone reflux, for 24 h each. TpBd-COF was subjected to another Soxhlet extraction under methanol reflux for 24 h.

### 2.3. Characterization Techniques

Microwave-assisted COF syntheses were conducted using a CEM Discover SP instrument. Samples and mixtures were sonicated in a Bandelin Sonorex RK100H ultrasonic bath. Fourier-Transform Infrared (FTIR) spectroscopy was carried out on an Agilent Technologies Cary 630 FTIR spectrophotometer equipped with attenuated total reflectance (ATR). Spectra were collected between 650 and 4000 cm^–1^ at a resolution of 2 cm^–1^. Proton nuclear magnetic resonance (^1^H NMR) spectra were obtained using a Bruker AMX spectrophotometer, operating at 400 MHz, using tetramethylsilane (TMS) as an internal standard. Thermogravimetric analyses (TGA) were conducted on a Nietzsch Tarsus TG 209 thermal analyzer. Samples were placed on an Al_2_O_3_ crucible and heated from room temperature up to 600 °C at 10 °C min^–1^ under nitrogen atmosphere (50 cm^3^ min^–1^ flow). X-ray diffraction (XRD) data were collected at room temperature by a PANalytical X’Pert diffractometer, equipped with an X’Celerator detector and secondary monochromator in Bragg–Brentano geometry. The measurements were carried out using 40 kV and 30 mA, CuKα radiation (λ_α1_ = 1.54060 Å and λ_α2_ = 1.54443 Å), 0.017°/step, and 200 s/step, in a 4–40° 2θ angular range. The Brunauer–Emmett–Teller (BET) specific surface area was obtained through nitrogen adsorption (Micrometrics ASAP 2000). pH measurements were carried out using a PHM240 MeterLab (Radiometer Copenhagen) coupled with a pH conjugated electrode (WTW, Sentix 22). UV–Vis spectrophotometry was carried out on a Shimadzu UV-2600i spectrophotometer. Flame atomic absorption spectroscopy (FAAS) measurements were carried out on a Unicam Solaar 939 spectrometer. Samples subjected to characterization were previously lyophilized in a Labconco Freezone 4.5 instrument after nitrogen bath freezing. COF morphologies were analyzed by scanning electron microscopy (SEM) on a JEOL Model 5310, operating at 2 kV at low pressure. The samples were previously coated with a thin gold film.

### 2.4. Adsorption Experiments

Removal efficiency and sorption isotherm studies were performed following a procedure similar to those described elsewhere [53,54]. For sorption isotherm studies, small COF samples (ca., 10 mg) were added to 10 mL adsorbate solutions (25 ppm–300 ppm) in ultrapure water. The suspensions were incubated (120 rpm, 25 °C) for 72 h. Samples were then centrifuged at 4000 rpm for 1 h and the supernatant was collected for pollutant quantification. Tests were conducted in duplicate. For removal efficiency studies, 4.0 mg COF samples were added to 10 mL of dye (20 ppm) or heavy metal (5 ppm) adsorbate solutions.

For sorption kinetics measurements, 10.0 mg COF samples were added to 100 ppm adsorbate solutions (40 mL) in ultrapure water. The suspensions were incubated (120 rpm, 25 °C) and 500 μL samples were consecutively collected at different time periods (5 min to 96 h). The dye quantification was carried out by collecting the supernatant of samples after centrifugation at 4000 rpm for 1 h. MO sorption kinetics experiments were conducted using 20.0 mg adsorbent samples and 20 ppm adsorbate solutions (40 mL).

COF titration measurements were performed by adding 50.0 mg COF samples to 40 mL of ultrapure water, followed by titration with sodium hydroxide 7.65 mM solution standardized using KHP 5 mM solution.

COFs containing –SO_3_H groups were neutralized prior to sorption experiments. Solution pH was maintained at 6–7 for all experiments.

Dye quantification was carried out using UV–Vis spectrophotometry (UV-2600i, Shimadzu, Kyoto, Japan). For each calibration curve, up to 12 standard solutions were prepared from adsorbate stock solutions (100 ppm), in concentrations ranging from 0.1 to 25 ppm, in ultrapure water. LOD and LOQ were determined based on the measuring of blank solutions in triplicate. Statistical parameters are detailed in Appendix A.

Metal ions concentrations were measured by Flame Atomic Absorption Spectroscopy (FAAS) using the procedure described elsewhere [53].

In this study, Langmuir and Freundlich sorption isotherm models were considered to model sorption isotherms. The Langmuir model is a thermodynamically derived equation and assumes an “ideal” adsorbent: homogeneous surface, well-defined, energetically equivalent sorption sites, and sorption defined by a monolayer. The Langmuir isotherm is given by Equation (1):(1)$$q_{e}=\frac{q_{m}K_{L}C_{e}}{1+K_{L}C_{e}}$$
where $q_{e}$ is the adsorption capacity at equilibrium and $q_{m}$ is the maximum adsorption capacity (monolayer saturation). $C_{e}$ corresponds to the adsorbate concentration at equilibrium and $K_{L}$ is the Langmuir constant.

The Freundlich sorption model is empirically derived and can be described by Equation (2):(2)qe=KFCe1nF
where $K_{F}$ stands for the Freundlich constant and $n_{F}$ is the surface heterogeneity factor. When 1nF is superior to 1, the adsorption mechanism is described as cooperative, resulting from multilayer adsorption; when it is inferior to 1, the process is mainly controlled by chemisorption [55].

Sorption kinetics studies were also performed to evaluate adsorbate-adsorbent interactions as a function of time. The pseudo-first-order equation, also known as the Lagergen equation, can be described by Equation (3):(3)$$q_{t}=q_{e}\left(1-e^{-k_{1}t}\right).$$
where $k_{1}$ defines the first-order kinetic constant. The equation expresses the adsorption capacity $q_{t}$ for a given time t.

The pseudo-second-order equation, also known as the Ho and Mckay equation, is given by Equation (4):(4)$$q_{t}=\frac{k_{2}q_{e}^{2}t}{1+k_{2}q_{e}t}$$
where $k_{2}$ describes the second-order kinetic constant [55].

## 3. Results and Discussion

### 3.1. Synthesis of Monomers and COF Materials

COFs were prepared via condensation between TFP and nonfunctionalized 4-phenylenediamine (Pa), benzidine (Bd) and 4,4′-ethylenedianiline (Bba), C_2_ symmetric diamines, as well as functionalized benzidine derivatives, 2,2′-dinitro- [1,1′-biphenyl]-4,4′-diamine [Bd(NO_2_)]_2_ and 4,4′-diamine- [1,1′-biphenyl]-2,2′-disulfonic acid [Bd(SO_3_H)]_2_. A scheme of the synthesis routes is shown in Figure 1 and the compounds are detailed in Table 1.

**Table 1 polymers-14-03096-t001:** Synthesis parameters of prepared TFP-based β-ketoenamine COFs.

Name	Diamine (s)	Molar Ratio (mol/mol)	Yield ^1^
TpPa-COF	Pa		96%/--
TpBd-COF	Bd		80%/--
TpBba-COF	Bba		80%/--
TpBd(NO_2_)_2_-COF [50%]	Bd/Bd(NO_2_)_2_	50/50	65%/64%
TpBd(NO_2_)_2_-COF [100%]	Bd(NO_2_)_2_		20%/53%
TpBd(SO_3_H)_2_-COF [50%]	Bd/Bd(SO_3_H)_2_	50/50	52%/87%
TpBd(SO_3_H)_2_-COF [100%]	Bd(SO_3_H)_2_		9%/77%

^1^ (microwave/solvothermal).

### 3.2. Characterization and Adsorption Studies of Nonfunctionalized COFs

In order to elucidate on the formation of new linkage types, nonfunctionalized COFs were analyzed via FTIR. The spectrum of TpPa-COF (Figure 2a, black line) was compared to those of its corresponding building monomers: TFP (Figure 2a, purple line) and 4-phenylenediamine-Pa (Figure 2a, blue line). This analysis allowed us to infer that the bands due to the C–H aldehyde stretching (2883 cm^–1^) and N–H primary amine stretching (3375–3186 cm^–1^) vibration modes, characteristic of TFP and Pa, respectively, are not present in the TpPa-COF spectrum, confirming the occurrence of the condensation reaction. Moreover, the presence in the spectrum of TpPa-COF (Figure 2a, black line) of intense bands attributed to the C=C stretching (1578 cm^–1^), C=O stretching (1600 cm^–1^), and C–N stretching (1253 cm^–1^) modes, and the absence of vibration modes due to the O–H stretching band (ca. 3000 cm^–1^) and to the C=N stretching band (around 1650 cm^–1^), corroborate the occurrence of the β-ketoenamine form of the COF. It should be stressed that both C=C and C=O bands appear at lower wavenumbers than expected probably on account of the extensive electronic conjugation of the COF, and its crystalline nature [32]. Additionally, other COF bands emerge at 827 and 995 cm^–1^, due to the *sp^2^* carbon C–H bending mode, and at 1520 cm^–1^, possibly due to the N–H bending mode [32]. Moreover, the FTIR spectra of the nonfunctionalized TpBd-COF (Figure 2b, brown line) and TpBba-COF (Figure 2b, orange line) samples proved to be very similar to TpPa-COF, with the aforementioned C=C, C=O, and C–N stretching bands, indicative of the β-ketoenamine tautomerization, appearing at the same wavenumber range.

The thermal stability of the COFs was evaluated by TGA. The TGA curves and the corresponding first-derivative curves of the nonfunctionalized TpPa-COF, TpBd-COF, and TpBba-COF are shown in Figure 3. Comparison of the three curves allows the conclusion that the thermal degradation of all three materials occurs essentially in two steps, suggesting the breaking of the same type of chemical bonds and polymer interlayer interactions. The first degradation event occurs at 323 and 314 °C, for TpBd-COF and TpBba-COF, respectively. For TpPa-COF, however, the same degradation event happens at a significantly higher temperature (340 °C). Given that the linkage types are the same among all samples, the latter result strongly suggests a more robust interlayer packing and more favorable, higher-energy π orbital stacking interactions. In contrast, the lower stability observed for TpBba-COF may be correlated with the conformational freedom introduced in the reticulated structure by the presence of 4,4′-ethylenedianiline (Bba) as the starting monomer. Bba contains *sp^3^*-hybridized carbons in its backbone, unlike the other diamines used, thus making the structure less thermally resistant overall. The second degradation event occurs at 462, 470, and 469 °C for TpPa-COF, TpBd-COF, and TpBba-COF, respectively. The proximity of these values once again suggests the structural similarities among the COF samples. Interestingly, the lowest degradation temperature was observed for TpPa-COF, which may indicate that the compound was the most significantly affected by the structural alterations introduced by the first degradation phenomenon [32,52,56].

Specific surface areas for the nonfunctionalized COFs were obtained by nitrogen (N_2_) adsorption and BET analysis. Figure 4 shows a representative N_2_ adsorption/desorption isotherm onto nonfunctionalized COFs. It can be observed that the N_2_ adsorption follows a type II isotherm with a slight H1 hysteresis for all three polymers [57]. This evidence suggests that all three COFs are mesoporous materials. In fact, by applying the BET method, we can find that the average pore diameter ranges from 7.6 to 20.8 nm, for TpPa-COF and TpBd-COF, respectively (see Table 2). However, it should be noted that the effect of diamine size on COFs follows a different trend when the surface area is evaluated. In this case, the COF with the highest surface area—TpPa-COF—yields the lowest pore size.

BET surface area measurements suggest that the starting monomers of larger dimensions led to a less available surface area. TpPa-COF, prepared using the smallest diamine (Pa), exhibited the highest result (83 ± 2 m^2^ g^–1^), whereas TpBd-COF and TpBba-COF, built from larger diamines, exhibited smaller surface areas (69 ± 1 and 32.4 ± 0.8 m^2^ g^–1^, respectively). It is also interesting to note that there seems to be a correlation between the average pore size and the size of the starting diamine used, with a lengthier building block leading to bigger pore dimensions. However, for TpBba-COF, formed from the diamine with the largest dimensions, the lowest BET surface area and a relatively small average pore size (9.0 nm) were observed. Again, this may be due to the added conformational freedom introduced by the –CH_2_–CH_2_– linkages from Bba, resulting in a collapsed porous structure with a less accessible surface area.

The XRD patterns obtained for TpPa-COF, TpBba-COF, and TpBd-COF are shown in Figure 5.

In the case of TpPa-COF, the most prominent peaks were observed at 4.8, 26.6, and 27.5° (Figure 5, black line). The 4.8 and 27.5° reflections were attributed to the reflections of the (100) and (001) planes, respectively [32]. Less intense peaks were detected at 8.8, 11.8, 12.8, and 14.7° (Figure 5, black line). For TpBd-COF, a very strong, sharp peak was found at 5.9°, with less intense peaks emerging at 11.2, 13.2, and 27.5° (Figure 5, blue line). Both these COFs appear to be more ordered than TpBba-COF. In fact, their XRD patterns show better-defined peaks than those seen in the XRD pattern of TpBba-COF (Figure 5, red line). The latter sample produced an ill-defined intense peak at 5.6° and minor peaks peaking at 13.2, 26.2, and 28.4°. This difference in crystallinity may be due, in part, to the *sp^3^* carbon present in the Bba moiety, resulting in less rigid and, therefore, less planar 2D layers. The above XRD data obtained are in perfect agreement with the results discussed in the literature for TpPa-COF [32], TpBba-COF [56], and TpBd-COF [52].

The surface morphologies of the nonfunctionalized COF materials were further evaluated by SEM. Figure 6 shows the SEM images obtained for TpPa-COF, TpBba-COF, and TpBd-COF.

The texture of TpPa-COF (Figure 6a) is characterized by small, homogeneous spherical aggregates (2–3 μm), sprouting from other structures of equivalent size, indicating that the growth process occurred by formation of independent polymer nuclei. The observed surface roughness is also in agreement with the high porosity, and the corresponding surface area of the material [32]. TpBba-COF (Figure 6b) shows varying morphology patterns, a result that suggests structural heterogeneity and mesoporosity. Different shapes are observed (from spherical and tubular to sheet-like aggregates) of varying sizes (up to 10 μm), with no apparent order. Morphology patterns for both COFs agree with the obtained BET surface areas: surface roughness and smaller average aggregate sizes lead to a higher available surface area. TpBd-COF (Figure 6c) exhibits irregular aggregates of varying shapes and dimensions, with a highly disordered and porous surface.

The performance of nonfunctionalized COFs for the adsorption of two dyes with different charges, methylene blue (MB, cationic) and methyl orange (MO, anionic), was assessed. Based on preliminary results (not shown) and for a selective evaluation of all adsorbents, a solid/liquid ratio equal to 0.4 mg mL^−1^ was chosen. Figure 7 shows the removal efficiencies for the adsorption of the two dyes on TpPa-COF, TpBba-COF, and TpBd-COF.

TpBd-COF showed the highest removal efficiency from among the nonfunctionalized COFs, with a removal of approximately 70% toward MB. It was observed that the removal efficiencies for MO are significantly lower than those obtained for MB. Moreover, a linear relationship between the average COF pore diameters and dye removal percentage is observed; i.e., the removal efficiency increases by increasing the pore diameters of materials. This can be explained by the easier access of the pollutant to the polymer backbone by materials with wider pore lengths.

The effect of MB adsorption on the COF structure was also evaluated. SEM analyses confirmed the modification of the surface morphology of nonfunctionalized COFs after adsorption (Figure 8).

Pre-adsorption TpPa-COF (Figure 8a1) exhibits a “flower-like” morphology, with spherical aggregates crystallizing independently and growing “petals” from their centers. These petal-like shapes act as an anchor to redirect polymer growth [32]. After adsorption (Figure 8a2), the surface is significantly eroded, becoming, in general, more diffuse and less compact. A similar effect can be observed for the SEM images of TpBd-COF (Figure 8c1,c2). Regarding TpBba-COF (Figure 8b1,b2), no significant morphological changes seemed to occur upon adsorption.

Moreover, the TGA curve recorded for the TpPa-COF sample post-adsorption also confirms the existence of trapped MB adsorbate (Appendix A), in particular, at 280 °C, where a slight mass loss is observed, due to MB degradation. The highest mass loss percentage also suggests that the adsorption process leads to a loss of thermal stability, which is in agreement with the loss of superficial integrity observed in SEM.

Nonfunctionalized COFs were further used as adsorbents for MB isotherm sorption studies. Two equations (Langmuir—Equation (1), and Freundlich—Equation (2)) were fitted to experimental values, and the fitting parameters are reported in Table 3.

In general, the isotherm curves fitted the Langmuir model better, which suggests a tendentially homogeneous surface and a predominantly monolayer-controlled adsorption. The highest MB uptake was observed for TpBd-COF, with a maximum adsorption capacity of (36 ± 1) mg g^–1^, according to the Langmuir model. Sorption kinetics fitting parameters can be found in Table 4.

The sorption experiments in general were better described by the pseudo-second-order equation. This suggests that the adsorption process is mainly due to chemisorption, which means that chemical reactions between the adsorbate and adsorbent are taking place at the interface. It also suggests monolayer adsorption. Sorption isotherms and kinetics are detailed in Figure 9.

### 3.3. Characterization and Adsorption Studies of Functionalized TpBd-Based COFs

TpBd-based functionalized COFs bearing –NO_2_ and –SO_3_H groups were prepared and characterized. Figure 10 shows the FTIR spectra for the functionalized COFs, TpBd(NO_2_)_2_-COF [100%] and TpBd(SO_3_H)_2_-COF [100%], compared to the spectra of their corresponding starting diamines. As discussed in Figure 2, the spectra of the COFs are characterized by the absence of diamine N–H stretching bands (3474–3360 cm^–1^) (dark blue line and dark green line in Figure 10, respectively). Comparing the nitrated COF and the corresponding starting diamine, the presence of the aromatic N–O symmetric and asymmetric stretching bands at 1337 cm^–1^and 1508 cm^–1^, respectively, indicate the presence of –NO_2_ groups in the COF structure [58]. Concerning the sulfonated COF (Figure 10, dark green line) and its diamine (light green line—Figure 10), both produce a set of bands ranging from 1247 to 1100 cm^–1^ due to the S=O stretching vibration mode. The absence of the wide O–H stretching band in the COF spectrum, present in the diamine spectrum due to intramolecular hydrogen bonding, suggests an ordered polymer structure where pore wall functional groups are not mutually accessible [49].

In order to evaluate the influence of the functionalization on the COFs’ thermal stabilities, the TGA curves of TpBd-COF and its nitrated and sulfonated derivatives were recorded and compared (Figure 11). Interestingly, the results seem to suggest that a larger functionalization ratio (mol/mol) leads to an increase in thermal stability. The pattern found for TpBd(NO_2_)_2_-COF [50%] (Figure 11a, light blue line) is very similar to that of the nonfunctionalized counterpart TpBd-COF (Figure 11a, brown line), with degradation steps occurring at 324 and 466 °C, respectively. For fully functionalized TpBd(NO_2_)_2_-COF [100%] (Figure 11a, dark blue line), the first event occurs at a much higher temperature (360 °C), resulting in a more extensive mass loss, and a second event is not observed up to 600 °C, the maximum temperature analyzed. A possible explanation is the polarization induced by the –NO_2_ groups on the polymeric resonance structure, thus increasing interlayer interaction energies and consequently thermal stability. Similar conclusions can be drawn from the comparison between TpBd-COF and sulfonated COF derivatives (Figure 11b).

A potentiometric acid–base titration was carried out to determine the pK_a_ values of sulfonated COFs (Appendix A). For TpBd(SO_3_H)_2_-COF [100%], a mass percentage of the incorporated Bd(SO_3_H)_2_ monomer of 70.1% was determined (for 50.0 mg of COF sample), similar to the theoretical value (71.1%). Moreover, a pK_a_ of 3.88 was estimated based on the Henderson–Hasselbalch equation. For the TpBd(SO_3_H)_2_-COF [50%] sample, the Bd(SO_3_H)_2_ mass percentage of the incorporated monomer was only estimated at 22.9%, half of the expected amount (45.6%). This suggests that, during the synthesis and crystallization processes, nonfunctionalized benzidine was selectively integrated in the structure, in detriment of its sulfonated derivative. This can be attributed to the overall lower reactivity of Bd(SO_3_H)_2_: the bulky, electron-withdrawing –SO_3_H groups make the diamine less nucleophilic and contribute to steric hindrance. A pK_a_ of 4.48 was obtained for the COF.

Functionalized COFs were used as adsorbents for the removal of MB and MO (Figure 12). Removal efficiencies ranging from 73% to 96% were obtained, with –SO_3_H-substituted COFs exhibiting the higher adsorption. The capture of MB, an anionic dye, increased with the degree of functionalization, whereas for MO, which is negatively charged at pH 6–7, a higher ratio of COF functionalization resulted in decreased removal efficiencies. These results point out that electrostatic interactions play a major role on the adsorption process [59].

To have a deeper understanding on the mechanism of MB sorption onto functionalized TpBd-COF materials, the sorption isotherms and kinetics were evaluated and data are shown in Figure 13.

The sorption isotherms were fitted by Langmuir and Freundlich equations, and data are shown in Table 5. It is interesting to note that functionalized TpBd-COFs were more accurately described by the Freundlich model. This confirms the surface heterogeneity induced by the pore wall functionalization and the consequences it has on adsorption. The high $n_{F}$ Freundlich parameters found additionally suggest chemisorption dynamics. The highest adsorption was determined for sulfonated TpBd(SO_3_H)_2_-COF [100%], with a Langmuir maximum adsorption capacity of 166 ± 13 mg g^–1^, much higher than those obtained for the corresponding nitro counterparts (48 to 84 mg g^–1^). In both cases, the sorption suggests the domination by chemisorption, in agreement with the pseudo-second-order best fitting to kinetics sorption data—Table 6.

Physical properties and respective MB maximum adsorption capacities for similar COFs and related polymers found in the literature are described in Table 7 and compared to the results observed in this work for TpBd(SO_3_H)_2_-COF [100%]. An overall comparison between the properties of the various adsorbents suggests that the presence of anionic groups within the porous structure is one of the key contributing factors for a high adsorption efficiency of MB, with the BET surface area of the polymer also representing an important, but less decisive parameter. For instance, TpBd(SO_3_H)_2_-COF [100%] exhibits a higher MB adsorption capacity than COFs with a much higher BET surface area but without anionic functionalization. In fact, the materials bearing cationic and uncharged groups exhibited the lowest adsorption results from all adsorbents [47,49,60,61]. A similar conclusion can be drawn for all other materials containing sulfonic or sulfonate groups within the porous structure [49,61]. Regarding nonfunctionalized polymers, a higher BET surface area seems to correlate, overall, with a higher removal efficiency. It is also worth noting the significant increase in efficiency reported by Dang and coworkers for TpBd(SO_3_^–^) [49]. The COF is analogous to the TpBd(SO_3_H)_2_-COF [100%] studied in this work—however, imidazolium salts were grafted onto the structure, and adsorption was tested at pH 9. Both modifications may explain the sharp contrast in the adsorption results reported.

In order to have further confirmation of the effect of electrostatic interactions on the adsorption process, the adsorption of different heavy metal ions (Cu(II), Pb(II), Ni(II), and Cd(II)) onto TpBd(NO_2_)_2_-COF [100%] and TpBd(SO_3_H)_2_-COF [100%], at pH 7, was further tested and data on removal efficiencies are shown in Figure 14. From the analysis of the figure, it can be concluded that the strongest acid polymer showed a much higher efficiency overall than the nitro-substituted counterpart. Cu(II) removal percentages were similar for both materials, while Ni(II) and Pb(II) adsorption was much more significant for TpBd(SO_3_H)_2_-COF [100%], suggesting a higher affinity of sulfonate-bearing surfaces toward the metals. The lower results observed for the –NO_2_ substituted COF may be attributed to the positive charge present in this group, partially repelling the metal ions. It is also relevant to note that, for both COFs, removal was the lowest for Ni(II) and Cd(II), possibly due to the higher ionic radii and, thus, lower charge densities.

## 4. Conclusions

In this study, a series of TFP-based β-ketoenamine-type COFs with varying degrees of –NO_2_ and –SO_3_H functionalization were synthesized, for pollutant adsorption in aqueous solution. FTIR analysis confirmed the linkage type of the final COFs and TGA showed that higher functionalization percentages lead to higher thermal stability. N_2_ sorption confirmed the mesoporosity of the solids, while SEM analysis revealed that there were significant modifications in the COF surfaces due to the MB adsorption process. The titration of –SO_3_H-functionalized COFs showed a clear selectivity process during COF crystallization. Adsorption studies confirmed a much higher affinity for COFs functionalized with electron-withdrawing groups toward cationic adsorbates, with a maximum adsorption capacity as high as 166 ± 13 mg g^–1^ for TpBd(SO_3_H)_2_-COF [100%] toward MB adsorption, according to the Langmuir isotherm model. Sorption kinetics further showed the positive influence of pore wall functionalization on the sorption dynamics. The sulfonated COFs also proved to be generally better adsorbents of metal species, compared to the nitrated counterparts, revealing the selectivity in electrostatic interactions. These results confirm the high potential of COF as highly practical, highly tunable materials for use in selective and efficient applications for adsorption processes.

## Figures and Tables

**Figure 1 polymers-14-03096-f001:**
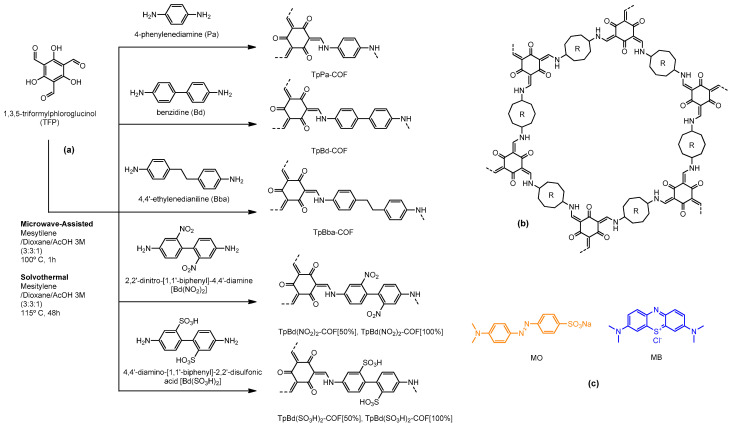
Scheme of the TFP-based β-ketoenamine COF syntheses (**a**), schematic structure of resulting 2D COF layer (**b**), and structures of the dyes used in adsorption studies (**c**).

**Figure 2 polymers-14-03096-f002:**
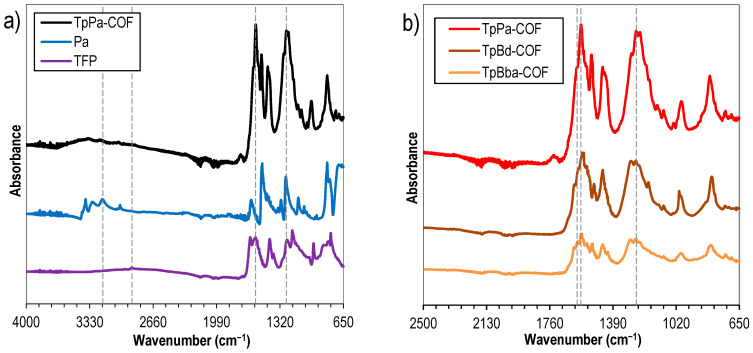
FTIR spectra of (**a**) TpPa-COF (black) and corresponding monomers TFP (purple) and Pa (blue); (**b**) TpPa-COF (red), TpBd-COF (brown), and TpBba-COF (orange).

**Figure 3 polymers-14-03096-f003:**
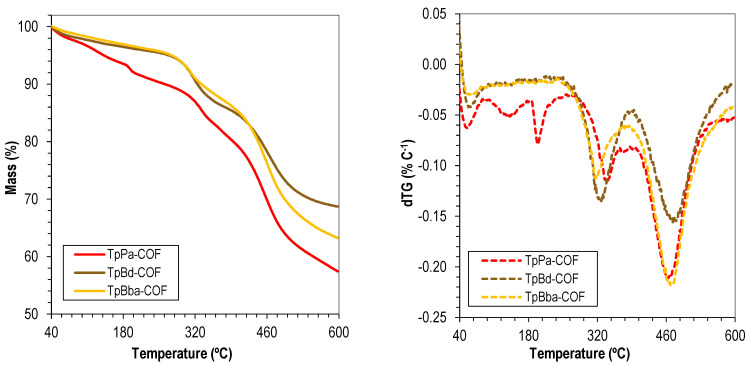
Thermograms (solid) and dTG (dashed) of TpPa-COF (red), TpBd-COF (brown), and TpBba-COF (orange).

**Figure 4 polymers-14-03096-f004:**
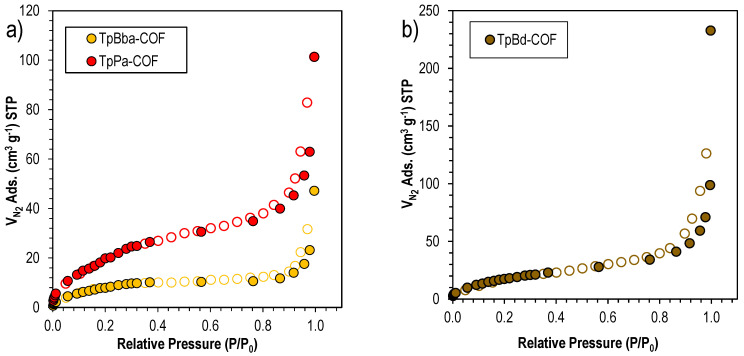
N_2_ sorption isotherms for (**a**) TpPa-COF (red) and TpBba-COF (orange), and (**b**) TpBd-COF (brown).

**Figure 5 polymers-14-03096-f005:**
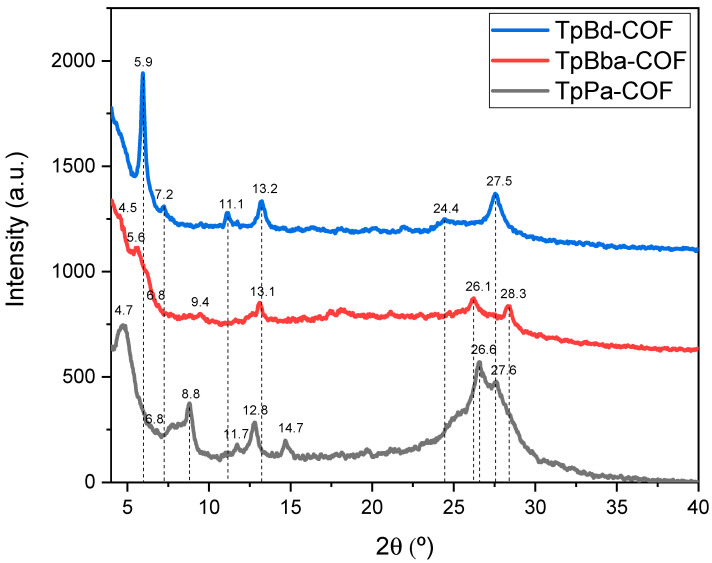
XRD patterns for TpPa-COF (black), TpBd-COF (blue), and TpBba-COF (red).

**Figure 6 polymers-14-03096-f006:**
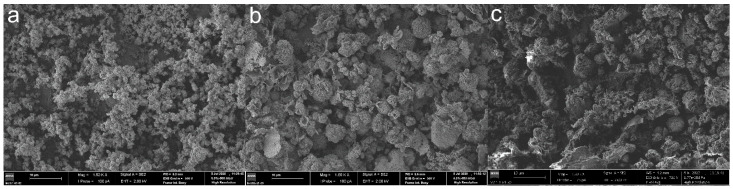
SEM images of TpPa-COF (**a**), TpBba-COF (**b**), and TpBd-COF (**c**) (×1500).

**Figure 7 polymers-14-03096-f007:**
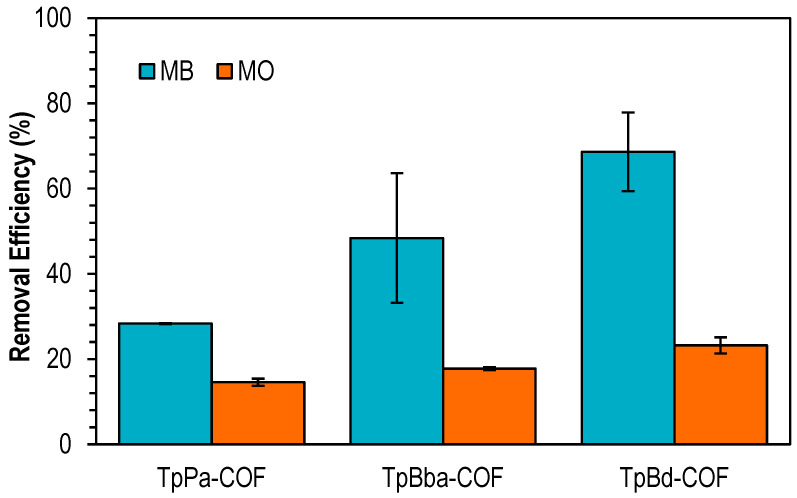
Removal efficiencies for the adsorption of MB (blue) and MO (orange) onto TpPa-COF, TpBba-COF, and TpBd-COF. Initial concentration of dyes: 20 ppm; mass of adsorbent: 4 mg.

**Figure 8 polymers-14-03096-f008:**
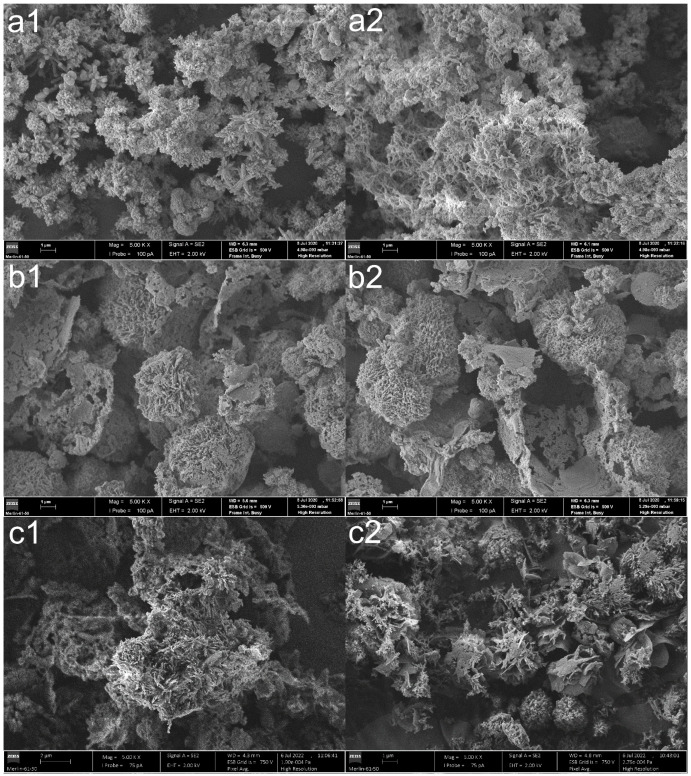
SEM images of (**a1**,**a2**) TpPa-COF, (**b1**,**b2**) TpBba-COF, and (**c1**,**c2**) TpBd-COF before (1) and after (2) MB adsorption (×5000).

**Figure 9 polymers-14-03096-f009:**
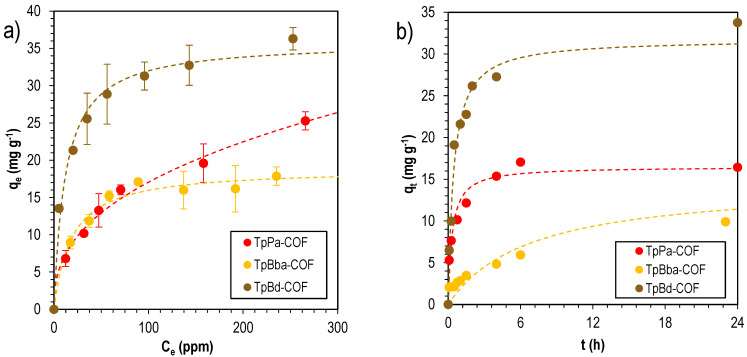
MB sorption isotherms (**a**) fitted to the Freundlich sorption model, and kinetics (**b**), fitted to the pseudo-second-order model, for TpPa-COF (red), TpBba-COF (orange), and TpBd-COF (brown).

**Figure 10 polymers-14-03096-f010:**
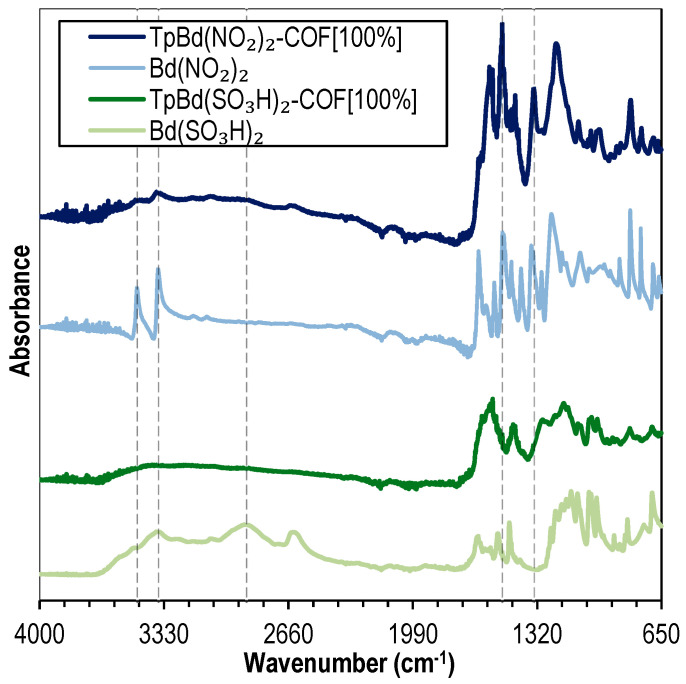
FTIR spectra of Bd(NO_2_)_2_ (light blue), TpBd(NO_2_)_2_-COF [100%] (dark blue), Bd(SO_3_H)_2_ (light green), and TpBd(SO_3_H)_2_-COF [100%] (dark green).

**Figure 11 polymers-14-03096-f011:**
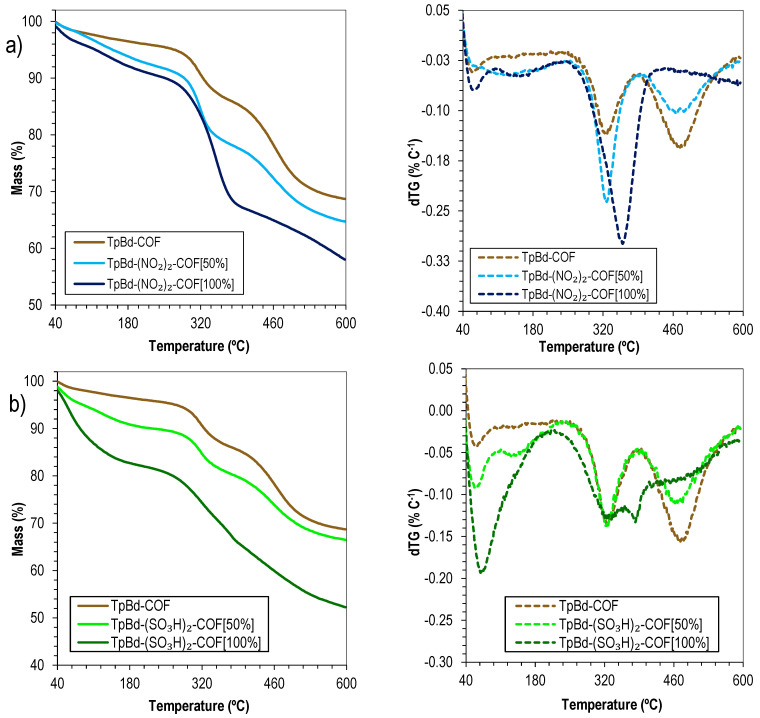
Thermograms (solid) and dTG (dashed) of (**a**) TpBd-COF (brown), TpBd(NO_2_)_2_-COF [50%] (light blue), and TpBd(NO_2_)_2_-COF [100%] (dark blue); (**b**) TpBd-COF (brown), TpBd(SO_3_H)_2_-COF [50%] (light green), and TpBd(SO_3_H)_2_-COF [100%] (dark green).

**Figure 12 polymers-14-03096-f012:**
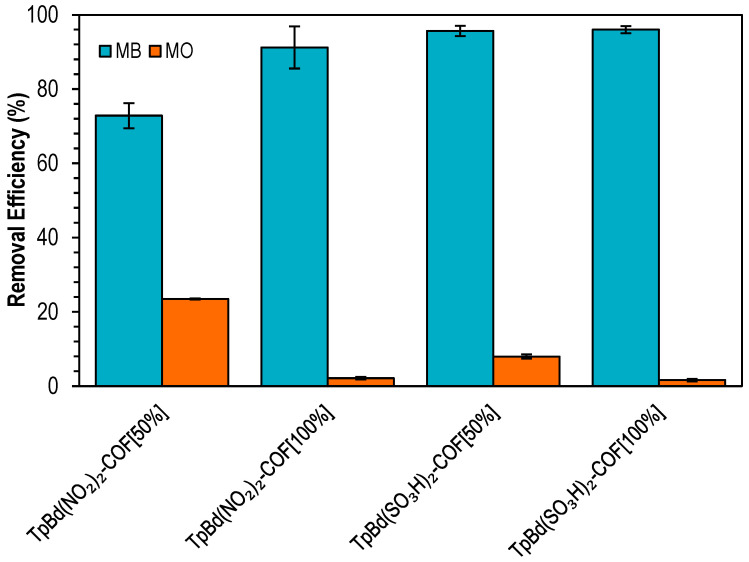
Removal efficiencies for the adsorption of MB (blue) and MO (orange) onto functionalized TpBd-COF materials. Initial concentration of dyes: 20 ppm; mass of adsorbent: 4 mg.

**Figure 13 polymers-14-03096-f013:**
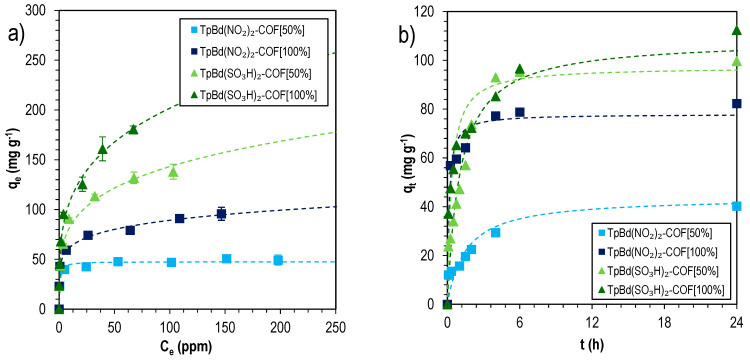
MB sorption isotherms (**a**), fitted to the Freundlich sorption model, and kinetics (**b**), fitted to the pseudo-second-order model, for functionalized COFs.

**Figure 14 polymers-14-03096-f014:**
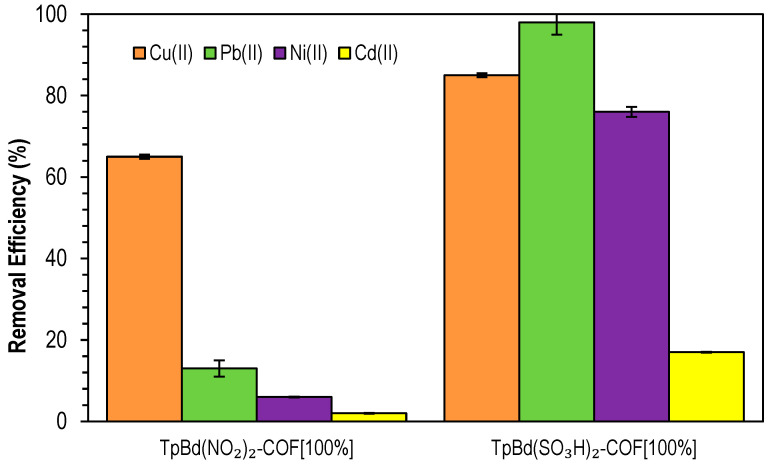
Removal efficiencies of Cu(II), Pb(II), Ni(II), and Cd(II) by functionalized TpBd COFs. Initial metal ion concentrations: 5 ppm at pH 4.

**Table 2 polymers-14-03096-t002:** N_2_ sorption isotherm parameters obtained for TpPa-COF, TpBd-COF, and TpBba-COF.

Name	BET Surface Area (m^2^ g^−1^)	Pore Volume (cm^3^ g^−1^)	Avg. Pore Diameter (nm)
TpPa-COF	83 ± 2	0.158	7.6
TpBd-COF	69 ± 1	0.359	20.8
TpBba-COF	32.4 ± 0.8	0.072	9.0

**Table 3 polymers-14-03096-t003:** Langmuir and Freundlich parameters for MB sorption isotherms on TpPa-COF, TpBba-COF, and TpBd-COF.

Name	Langmuir			Freundlich		
	*q_m_* (mg g^–1^)	*K_L_* (mg^–1^)	R^2^	*K_F_* (mg g^–1^)	*n_F_*	R^2^
TpPa-COF	29 ± 2	0.018 ± 0.003	0.9817	2.7 ± 0.3	2.5 ± 0.2	0.9722
TpBd-COF	36 ± 1	0.08 ± 0.01	0.9866	11 ± 1	4.5 ± 0.4	0.9854
TpBba-COF	18.9 ± 0.8	0.06 ± 0.01	0.9794	6 ± 1	5 ± 1	0.9511

**Table 4 polymers-14-03096-t004:** Pseudo-first-order and pseudo-second-order fitting parameters for sorption kinetics of MB for TpPa-COF, TpBba-COF, and TpBd-COF, at 25º C.

Name	Pseudo-First Order			Pseudo-Second Order		
	*q_e_* (mg g^−1^)	*k*_1_ (min^−1^)	R^2^	*q_e_* (mg g^−1^)	*k*_2_ (g mg^−1^ min^−1^)	R^2^
TpPa-COF	16 ± 1	0.03 ± 0.01	0.9023	16.5 ± 0.8	0.05 ± 0.01	0.9606
TpBd-COF	28 ± 1	0.030 ± 0.006	0.9389	32 ± 1	0.037 ± 0.006	0.9777
TpBba-COF	13 ± 1	0.0021 ± 0.0006	0.8702	15 ± 2	0.0025 ± 0.0008	0.9056

**Table 5 polymers-14-03096-t005:** Langmuir and Freundlich parameters for MB sorption isotherms for functionalized TpBd-COFs.

Name	Langmuir			Freundlich		
	*q_m_* (mg g^−1^)	*K_L_* (mg^−1^)	R^2^	*K_F_* (mg g^−1^)	*n_F_*	R^2^
TpBd(NO_2_)_2_-COF [50%]	48 ± 1	1.6 ± 0.3	0.9835	30 ± 2	10 ± 2	0.9724
TpBd(NO_2_)_2_-COF [100%]	84 ± 6	1.2 ± 0.7	0.8849	44 ± 1	6.5 ± 0.3	0.9966
TpBd(SO_3_H)_2_-COF [50%]	135 ± 7	0.28 ± 0.07	0.9646	48 ± 3	4.2 ± 0.3	0.9879
TpBd(SO_3_H)_2_-COF [100%]	166 ± 13	0.4 ± 0.1	0.938	59 ± 3	3.8 ± 0.2	0.9924

**Table 6 polymers-14-03096-t006:** Pseudo-first-order and pseudo-second-order fitting parameters for sorption kinetics of MB for different functionalized TpBd-COF materials, at 25 °C.

Name	Pseudo-First Order			Pseudo-Second Order		
	*q_e_* (mg g^−1^)	*k*_1_ (min^−1^)	R^2^	*q_e_* (mg g^−1^)	*k*_2_ (g mg^−1^ min^−1^)	R^2^
TpBd(NO_2_)_2_-COF [50%]	42 ± 4	0.007 ± 0.002	0.8261	44 ± 4	0.011 ± 0.004	0.8760
TpBd(NO_2_)_2_-COF [100%]	73 ± 4	0.09 ± 0.03	0.9224	78 ± 3	0.13 ± 0.05	0.9650
TpBd(SO_3_H)_2_-COF [50%]	97 ± 5	0.012 ± 0.002	0.9509	109 ± 6	0.016 ± 0.003	0.9573
TpBd(SO_3_H)_2_-COF [100%]	88 ± 7	0.04 ± 0.01	0.8078	97 ± 7	0.05 ± 0.02	0.9033

**Table 7 polymers-14-03096-t007:** Physical properties and MB adsorption parameters for other COFs and porous polymers found in the literature.

Adsorbent	Linkage Type	Pore Wall Functionalization	BET Surface Area (m^2^ g^−1^)	Pore Size (nm)	*q*_max_ (mg g^−1^)	Ref.
TS-COF-1	Imide	–	1484	3.1	1691	[62]
TS-COF-2	β-ketoenamine	–	756	1.1	377	[62]
TPT-DMBD-COF	Imine	CH_3_	279.5	4.0	45.5	[47]
CON-1 ^a^	C-C	–	26	2.5	– ^b^	[39]
SCF-FCOF-1	Imine	F	2056	3.5	108.9	[40]
bCOF	Imide	–	856	–	63.3	[60]
MOP-2 ^c^	Azine	–	327	–	1153	[63]
PPOP-SO_3_H	Enamine	SO_3_H	–	–	980.4	[61]
TpPa-SO_3_	β-ketoenamine	SO_3_^− d^	70.8	1.7	1806.9 ^e^	[49]
TpBd-(SO_3_)_2_	β-ketoenamine	SO_3_^− d^	60.5	9.3	2653.3 ^e^	[49]
TpBd-(SO_3_H)_2_	β-ketoenamine	SO_3_H	32.4 ± 0.8	9.0	166 ± 13	This Work

^a^: polycationic COF. ^b^: residual adsorption, value not reported. ^c^: Fe_3_O_4_ MNP—COF composite. ^d^: ionic liquid grafted. ^e^: evaluated at pH 9.

## Data Availability

No new data were created or analyzed in this study. Data sharing is not applicable to this article.

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
