# Peer review of "β-Ketoenamine Covalent Organic Frameworks—Effects of Functionalization on Pollutant Adsorption"

_polymers, 2022, doi:10.3390/polym14153096_

Round 1
Reviewer 1 Report
In general, the authors try to report three key COF materials, TpPa-COF, TpBd-COF & TpBba-COF. However, it does confuse me based on the materials characterization results, from figure 2 to figure 6. Every figure shows a quality comparison, but based on a different group of COFs. This would not work to support the authors' claims. Similar technical issues can be found on performance evaluation. Therefore, I would strongly recommend the authors to restructure the manuscript focusing a certain group of COFs for both materials characterization and performance evaluation.
Author Response
We really appreciated the Reviewer’s comments. In fact, we agree that our choice was not initially the best. Now, the discussion is divided in 3 different sections. They are:
3.1 Synthesis of Monomers and COF Materials
3.2. Characterization and Adsorption Studies of non-Functionalized COFs
3.3. Characterization and Adsorption Studies of Functionalized TpBd-based COFs
The text has not changed significantly despite the colour regions highlighted in the revised version; however, some data have been included in the manuscript to provide self-consistency.

Reviewer 2 Report
Some minor English corrections: P1 L30: chemical interactions; P1 L32: clearly demonstrating
Pages 5, 6, 7, 9, 10, 13, etc: Error! Reference source not found. Should be clarified! Please revise the references and cite them again.
Author Response
Some minor English corrections: P1 L30: chemical interactions; P1 L32: clearly demonstrating
We really apologize for such mistakes. English has been checked throughout the manuscript.
Pages 5, 6, 7, 9, 10, 13, etc: Error! Reference source not found. Should be clarified! Please revise the references and cite them again.
We really apologize for that. We have formatted citing Tables and Figures as a hyperlink and when converting to pdf that Error information appears. That was fixed.

Round 2
Reviewer 1 Report
The authors have addressed the reviewers' comments. Therefore, I recommend it to be accepted as it is.